# Performance of Culture Using a Semi-Automatic Needle as a Novel Tool for Collecting Lymph Node Samples for the Diagnosis of Canine Visceral Leishmaniasis

**DOI:** 10.3390/ani15010107

**Published:** 2025-01-05

**Authors:** Artur Augusto Velho Mendes Júnior, Fabiano Borges Figueiredo, Luiz Cláudio Ferreira, Lucas Keidel, Renato Orsini Ornellas, Adilson Benedito Almeida, Fernanda Nunes Santos, Luciana de Freitas Campos Miranda, Andreza Pain Marcelino, Sandro Antonio Pereira, Rodrigo Caldas Menezes

**Affiliations:** 1Carlos Chagas Institute, Oswaldo Cruz Foundation, Rua Prof. Algacyr Munhoz Mader, 3775, Curitiba 81350-010, Brazil; aavelho@gmail.com (A.A.V.M.J.); fabiano.figueiredo@fiocruz.br (F.B.F.); 2Laboratory of Clinical Research on Dermatozoonoses in Domestic Animals, Evandro Chagas National Institute of Infectious Diseases, Oswaldo Cruz Foundation, Av. Brasil, 4365, Rio de Janeiro 21040-360, Brazil; cowboylcf321@gmail.com (L.C.F.); lucas.keidel@ini.fiocruz.br (L.K.); renatoorsini@yahoo.com.br (R.O.O.); adilson.almeida@ini.fiocruz.br (A.B.A.); sandro.pereira@ini.fiocruz.br (S.A.P.); 3Laboratory of Clinical Research and Surveillance in Leishmaniasis, Evandro Chagas National Institute of Infectious Diseases, Oswaldo Cruz Foundation, Av. Brasil, 4365, Rio de Janeiro 21040-360, Brazil; santosfn.vet@gmail.com (F.N.S.); luciana.freitas@ini.fiocruz.br (L.d.F.C.M.); andreza.marcelino@ini.fiocruz.br (A.P.M.)

**Keywords:** canine visceral leishmaniasis, semi-automatic needle, parasitological diagnosis, popliteal lymph node, dogs

## Abstract

Visceral leishmaniasis caused by the protozoan *Leishmania* (*Leishmania*) *infantum* is a disease transmitted between dogs and humans. The current lymph node sampling techniques are unable to provide sufficient amounts of samples for laboratory tests. We, therefore, evaluated semi-automatic needle puncture as a novel tool for collecting lymph node samples for the diagnosis of canine visceral leishmaniasis. This technique uses a thick needle to increase the amount of sample and was compared with samples collected from the same lymph node by aspiration using a fine needle and by necropsy. Lymph node samples from 30 dogs seropositive for *L. infantum* were processed for use in the following laboratory tests: culture, immunohistochemistry, and histopathology. *Leishmania* spp. was detected by immunohistochemistry in 70% of the samples, and by histopathology in 33%. Culture positivity was 77% for the samples obtained by necropsy (87% in the first week), 73% for the samples obtained by fine needle aspiration (82% in the first week), and 63% for the samples obtained using semi-automatic needle puncture (95% in the first week). The combination of semi-automatic needle puncture and culture is an alternative for the detection of *Leishmania* spp. in the lymph nodes of dogs because of its efficiency, practicality, and fast results.

## 1. Introduction

Visceral leishmaniasis caused by *Leishmania* (*Leishmania*) *infantum* can affect humans and different animal species. It is mainly transmitted by vectors, and females of the sandfly species *Lutzomyia longipalpis* and *Lutzomyia cruzi* are the most common vectors in Brazil [1,2]. In the Americas, visceral leishmaniasis is a zoonosis and dogs are the main domestic reservoir; these animals are, therefore, one of the pillars of the disease control program in Brazil [3]. The clinical diagnosis is difficult and laboratory tests are, thus, necessary for confirmation.

Serological tests and clinical examination are usually the first step in the diagnosis of canine visceral leishmaniasis (CVL); however, parasitological tests are the reference standard for confirming infection with *Leishmania* spp. since they allow the demonstration of the parasite [1,4]. The most useful parasitological tests to confirm infection with *Leishmania* spp. during the routine diagnosis of CVL are cytopathological examination, culture, histopathology, and immunohistochemistry. Cytopathological examination permits us to detect amastigote forms in cytological samples and is a practical, rapid, low-cost, and noninvasive sampling method; however, it does not allow the identification of *Leishmania* to species level and has limited sensitivity ranging from 32% to 76% [5]. The culture of *Leishmania* sp., combined with the isoenzyme characterization of the species by electrophoresis, is 100% specific, with sensitivity ranging from 78.7% to 80.3% [6]. Therefore, culture is indicated for identifying and monitoring species that circulate in endemic areas [7]. Histopathology is important for the diagnosis of leishmaniasis since it permits us to identify amastigote forms in tissues and to concomitantly associate parasitism with histological lesions [8]. The immunohistochemistry of histological sections enables the visualization of amastigote forms under an optical microscope based on antigen–antibody reactions. Additionally, its diagnostic sensitivity in detecting *Leishmania* sp. in dogs [9] and human tissues [10] is greater than that of histopathology.

One limitation of parasitological tests is that the collection of clinical samples is often invasive and difficult, posing risks to the animal. Therefore, the type of tissue used for the confirmation of *L. infantum* infection is a critical aspect that needs to be addressed. The popliteal lymph node and spleen are the most recommended specimens for detecting *L. infantum* because they provide high positivity rates (72 to 83%) of *L. infantum* in parasitological tests, higher than the rates reported for bone marrow and skin [11]. Additionally, the popliteal lymph node is easily accessible when compared to other lymph nodes and sample collection is less invasive than that of spleen and bone marrow.

The main lymph node sampling techniques for the diagnosis of CVL are fine needle aspiration puncture (FNAP) and non-aspiration puncture because they are easy to use and good quality material can be obtained for examination, with few risks inherent to the technique. In addition, tissue aggressiveness is low and these techniques are less painful [12,13]. However, there are limitations such as the need for professional experience and the difficulty in obtaining a sufficient amount of sample for diagnosis when the lymph nodes are small and difficult to palpate. Furthermore, low cellularity, cell lysis, and blood contamination can compromise the cytological diagnosis by FNAP [12]. Collecting a lymph node fragment at necropsy or by excisional biopsy (the removal of the entire lymph node), by incisional biopsy (the removal of a lymph node fragment), or by puncture with a thick needle would solve the limitation of FNAP in terms of sample quantity, especially in the case of non-enlarged lymph nodes. Within this context, semi-automatic needle puncture (SANP) would permit to collect larger amounts of a lymph node sample than FNAP because of the use of a thick needle, thus increasing the accuracy of parasitological tests, as reported for the diagnosis of neoplasms [14]. Furthermore, SANP is a safe biopsy technique for the operator, more practical and faster than FNAP, and less invasive than incisional or excisional biopsy. The use of a semi-automatic needle for the puncture of lymph nodes and other organs is common in human medicine for diagnosing neoplasms [14] but has not yet been evaluated in dogs as a lymph node puncture technique for the diagnosis of CVL.

Evaluating an innovative approach to popliteal lymph node sampling for the parasitological diagnosis of CVL, the present study reports for the first time the performance of lymph node sampling by SANP compared to samples collected by FNAP and necropsy.

## 2. Materials and Methods

This cross-sectional study evaluating a diagnostic method used a convenience sample of 30 dogs selected during a serological survey carried out from 2018 to 2019 in the city of Barra Mansa, state of Rio de Janeiro, Brazil. All the dogs had positive results in the TR-DPP^®^ rapid immunochromatographic test (Bio-Manguinhos, Fiocruz, Rio de Janeiro, Brazil) and in the EIE-LVC^®^ enzyme immunoassay (Bio-Manguinhos, Fiocruz, Rio de Janeiro, Brazil), the official tests recommended by the Brazilian Ministry of Health for the diagnosis of CVL [15].

The animals were submitted for the clinical evaluation of specific clinical signs of CVL and then euthanized following the recommendations of the Ministry of Health for the control of CVL [15]. After the confirmation of death, the popliteal lymph node was located and manipulated until it was firm and prominent under the skin. The area was shaved and antisepsis was performed as follows: three applications per sampling method of gauze first soaked in 2% chlorhexidine, then in 10% povidone-iodine, and finally in 70% alcohol.

The first lymph node sample was collected by FNAP. Two drops (0.1 mL) were stored in a sterile saline solution containing 100 µg/mL fluorocytosine, 1200 IU/mL penicillin, and 1000 µg/mL streptomycin for 24 h and sent for parasite isolation by culture. The samples were incubated at 26–28 °C in Novy-MacNeal-Nicolle + Schneider’s biphasic media, supplemented with antibiotic and antifungal agents, and examined weekly for 30 days in order to identify promastigote forms [16]. After parasite isolation, the samples were expanded to produce parasite mass for subsequent isoenzyme characterization by multilocus enzyme electrophoresis (MLEE) [17]. In addition to the presence of promastigotes, the time necessary for parasite isolation and the rate of bacterial/fungal contamination of the technique were evaluated.

The second sample was collected by SANP (Figure 1) using a SemiCut semi-automatic disposable needle (16G or 1.3 mm in diameter and 15 cm in length) for soft tissue biopsy (MDL SRL, Delebio, Italy). The organ was perforated at a depth of about 1 to 2 cm, at an angle of 45°, using a special needle attached to an automatic spring-loaded pistol. A ribbon-shaped, approximately 3 to 5 mm long, fragment was removed through a shot, stored in sterile saline solution as described above, and sent for culture.

After the collection of the clinical samples by FNAP and SANP, a necropsy was performed to remove the entire organ. Two fragments were collected near the previous sampling sites. The first was fixed in 10% buffered formalin for the detection of amastigote forms of *Leishmania* spp. by histopathology and immunohistochemistry [18,19], and the second fragment was stored in a sterile saline solution containing antimicrobials and sent for culture.

The results of clinical evaluation and of the laboratory tests for the diagnosis of *Leishmania* spp. infection in dogs were analyzed descriptively using the Statistical Package for the Social Sciences (SPSS), version 16.0. Infection with *Leishmania* spp. was confirmed when the parasite was detected in at least one direct diagnostic test (culture, immunohistochemistry, and histopathology). The frequency of *Leishmania* spp. positivity in the various tissues using the different diagnostic techniques was analyzed descriptively.

## 3. Results

Twenty-six (86%) of the thirty animals studied showed clinical signs compatible with CVL. The most frequent clinical signs were splenomegaly (76%), dermatological changes (60%), lymphadenomegaly (40%), hepatomegaly (36%), and onychogryphosis (30%), in agreement with Solano-Gallego et al. [20]. A limitation of this study is that clinical staging of the dogs was not performed. All the *Leishmania* sp. isolates obtained by culture were characterized as *L. infantum* using the MLEE technique, a result expected for the endemic area investigated [21].

According to the data presented in Table 1, the frequencies of positivity by the culture of the necropsy samples (77%) were higher than those obtained for immunohistochemistry and histopathology (70% and 30%, respectively) (Figure 2) and agree with those reported in the literature [22]. However, an advantage of immunohistochemistry is that unlike culture, there are no losses due to microbiological contamination.

A comparison of the frequency of positivity between the different sampling techniques for culture showed that the necropsy provided the best result (77%), followed by FNAP (73%) and SANP (63%). However, the difference in the number of positive cases detected by the techniques was small, ranging from 1 to 4 cases. These results corroborate Furtado et al. [11] who reported a positivity of 72.3% and Mello et al. [22] who found a positivity of 73.8% in lymph node samples collected by necropsy from seropositive dogs from the same endemic region as the animals of the present study and examined by culture.

SANP and FNAP can be used for in vivo sampling. The present results demonstrate a good positivity rate of SANP. Furthermore, like FNAP, SANP can be performed in vivo and in the field for the diagnosis of CVL. The advantages of SANP over FNAP include more practical and faster sampling and that larger amounts of sample can be obtained, which can be used to perform complementary diagnostic techniques such as molecular biology, histopathology, and immunohistochemistry. However, SANP requires further studies regarding its invasiveness in dogs and the degree of tissue damage it causes in field studies with live animals. Within this context, one limitation of this study was the fact that SANP was conducted on dead animals, and variables such as tissue hemorrhage could, therefore, not be assessed. In addition, signs of pain were not analyzed. However, since this was the first time the technique had been evaluated, it was the most ethical approach.

The frequency of *L. infantum* positivity was higher when the animal had lymphadenopathy, regardless of the sampling technique (Table 2). One explanation for this result is that the increase in lymph node size is related to the inflammatory process caused by the increase in the number of parasites. In addition, a larger amount of tissue or aspirate can be obtained from an enlarged lymph node compared to normal-sized lymph nodes.

Although sensitive and 100% specific, a major disadvantage of culture for diagnosing *L. infantum* in dogs is the time necessary to obtain the final result, which ranges from 15 to 30 days. For CVL monitoring, especially in disease-free areas where the identification of the circulating *Leishmania* species is necessary, the interval between sample collection and the final result affects the adoption of surveillance and control measures. In this study, 95% of the samples collected by SANP were positive in the first week of culture (Table 3), a frequency slightly higher than that obtained for necropsy and FNAP. These findings agree with another study using different tissues [23].

Although our results demonstrate the efficiency of culture considering the positivity rate and time to diagnosis using the three sampling techniques (SANP, FNAP, and necropsy), this method has some limitations such as its dependence on the parasite load of the clinical sample and operational difficulties in collection and implementation that require training and an adequate infrastructure. Contamination or poor adaptation of the parasite to the environment can compromise the sensitivity of culture and underestimate the accuracy of other tests [11]. Despite satisfactory results in terms of the risk of microbiological contamination (<5%), SANP was the sampling technique with the highest contamination. One possible explanation for this result is the fact that the puncture needle guide, which is larger, is to a greater extent in direct contact with the skin than in FNAP. The skin harbors a wide variety of contaminating microorganisms that can interfere with the results of culture [24]. We, therefore, recommend enhancing skin antisepsis before sample collection by applying to the shaved skin at the puncture site at least three rounds of sterile gauze first soaked in 2% chlorhexidine, then in 10% povidone-iodine, and finally in 70% alcohol.

In this study, we observed a high frequency of positivity in lymph nodes regardless of the sampling technique. The lymph node may, therefore, be an organ of choice for parasitological tests, considering the lower degree of invasiveness when compared to skin, bone marrow, or spleen. The diagnosis of CVL is essential for routine veterinary medicine and for visceral leishmaniasis control and surveillance programs. Since there is no single technique that shows 100% sensitivity and specificity for diagnosing CVL, the combination of several diagnostic methods is the best approach to detecting this infection [25].

## 4. Conclusions

SANP of popliteal lymph nodes combined with culture can be used as an alternative tool for the diagnosis of *L. infantum* in dogs because of the high frequency of positivity, good specificity, and low rate of microbiological contamination. In addition, SNAP compared to FNAP and necropsy sampling is more practical, faster, and has a shorter time to positivity by culture. However, in vivo studies of lymph node sampling by SANP in dogs are needed to evaluate the occurrence of hemorrhage and the generation of discomfort to the animals associated with this technique.

## Figures and Tables

**Figure 1 animals-15-00107-f001:**
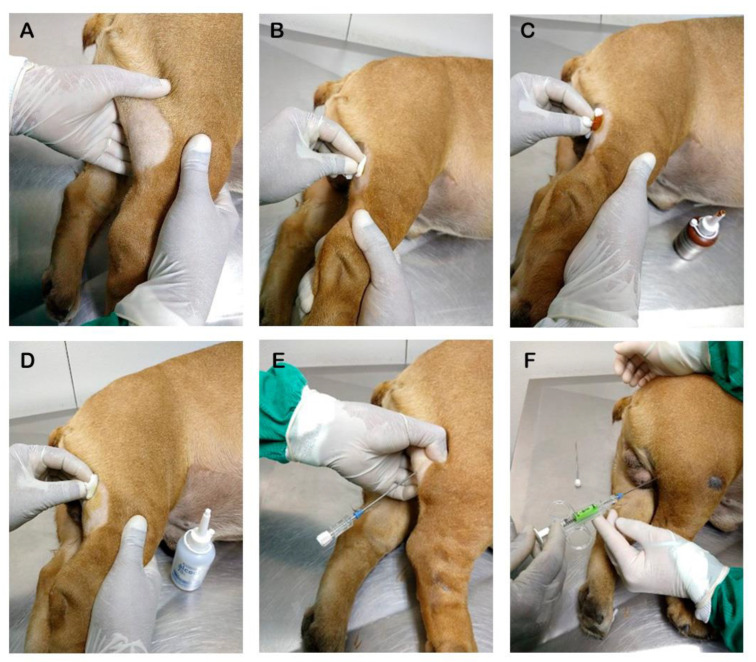
Schematic representation of popliteal lymph node sampling in a dog using a semi-automatic soft tissue needle. (**A**) Identification, shaving, and positioning of the popliteal lymph node. (**B**) Cleaning and antisepsis with 2% chlorhexidine. (**C**) Antisepsis using iodinated alcohol. (**D**) Antisepsis using 70% alcohol. (**E**) Attachment of the guide for semi-automatic needle sampling. (**F**) Needle inserted into the guide already fixed to the tissue and ready for collection after being fired by pressing the trigger.

**Figure 2 animals-15-00107-f002:**
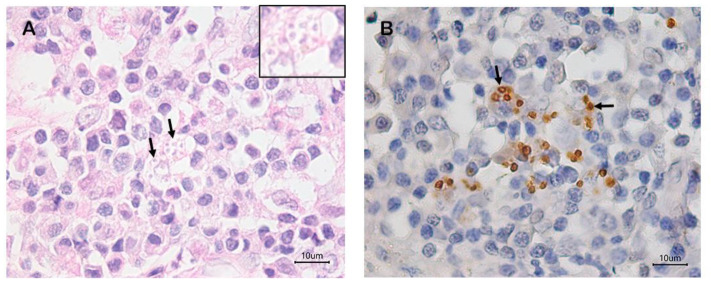
(**A**) Histological section of the popliteal lymph node of a dog showing amastigote forms of *Leishmania* sp. (arrows and inset) in the cytoplasm of macrophages. Hematoxylin–eosin; 100×objective. (**B**) Histological section of the popliteal lymph node of a dog showing brown-stained amastigote forms of *Leishmania* sp. (arrows) in the cytoplasm of macrophages. Immunohistochemistry; 40× objective.

**Table 1 animals-15-00107-t001:** Frequencies of *Leishmania infantum* positivity in popliteal lymph nodes of seropositive dogs sampled by semi-automatic needle puncture (SANP) and culture examination compared to sampling by fine needle aspiration (FNAP) and culture examination and sampling by necropsy and examination by culture, histopathology, and immunohistochemistry. Barra Mansa, Rio de Janeiro (2018–2019).

Sampling Technique + Parasitological Examination	Frequency of *Leishmania* Positivity in the Popliteal Lymph Node (N = 30)	
	N	%
Necropsy + culture	23	76.7
FNAP + culture	22	73.3
Necropsy + immunohistochemistry	21	70.0
SANP + culture	19	63.3
Necropsy + histopathology	10	33.3

N = number of dogs; % = percentage; FNAP = lymph node sample collected by fine needle aspiration puncture; SANP = lymph node sample collected by semi-automatic needle puncture.

**Table 2 animals-15-00107-t002:** Frequencies of *Leishmania infantum* positivity in the popliteal lymph nodes of the seropositive dogs sampled by semi-automatic needle puncture, fine needle aspiration puncture, and necropsy for culture according to the presence or absence of lymphadenomegaly. Barra Mansa, Rio de Janeiro (2018–2019).

Sampling Technique	Frequency of *Leishmania* sp. Positivity in Popliteal Lymph Nodes	
	Lymphadenomegaly (N = 12)	No Lymphadenomegaly (N = 18)
SANP	9 (75%)	10 (56%)
FNAP	9 (75%)	13 (72%)
Necropsy	11 (92%)	12 (67%)

SANP = semi-automatic needle puncture; FNAP = fine needle aspiration puncture.

**Table 3 animals-15-00107-t003:** Frequencies of *Leishmania infantum* positivity according to the time of culture of popliteal lymph node samples collected using SANP, FNAP, and necropsy. Barra Mansa, Rio de Janeiro (2018–2019).

Sampling Technique	Frequency of *Leishmania* sp. Positivity (N = 64) by Culture									
	1st Week		2nd Week		3rd Week		4th Week		Total	
	n	%	n	%	n	%	n	%	n	%
SANP	18	95	1	5	0	0	0	0	19	100
Necropsy	20	87	3	13	0	0	0	0	23	100
FNAP	18	82	2	9	1	4.5	1	4.5	22	100

N = total number of culture-positive samples considering all the sampling techniques; n = number of parasitological culture-positive samples per sampling technique.

## Data Availability

Data are contained within the article. The original contributions presented in the study are included in the article; further inquiries can be directed to the corresponding author.

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
