# Peer review of "Performance of Culture Using a Semi-Automatic Needle as a Novel Tool for Collecting Lymph Node Samples for the Diagnosis of Canine Visceral Leishmaniasis"

_animals, 2025, doi:10.3390/ani15010107_

Round 1
Reviewer 1 Report
Comments and Suggestions for Authors
Dear authors,
The novel technique+culture is not superior to traditional fine needle aspirate + culture technique, in terms of detection Leishmania positivity. But the methodology is well-designed and presented. When I read the manuscript, I could not understand, if the authors suggest to use this method. There is a major mis-conseption at the article in this point. At conclusion section, they should more clearly state.
The article also give frequency data in dogs, the title should involve this information in addition to assessment of novel technique. The technique also seems rather an alternative of fine needle aspirate.
1. There are many abbreviations in the text, you should use less for understanding and following the article. For example PC: culture , necroscopy: LN
2. Line 33. Correct “Leishmania (Leishmania) infantum “
3. Line 102. “To evaluate an innovative approach of popliteal lymph node sampling for the para- 102 sitological diagnosis of CVL, the present study reports for the first time the performance 103 of lymph node sampling by SANP when compared to samples collected by FNAP and 104 necropsy (LN) for the diagnosis of CVL by PC compared to HP and IHC as diagnostic 105 techniques” this sentence is not clear. Please revise it.
Comments on the Quality of English Language
Overall, the quality of English is well.
Author Response
Animals - Manuscript # 3344307
Response to reviewers
Reviewer 1
The novel technique+culture is not superior to traditional fine needle aspirate + culture technique, in terms of detection Leishmania positivity. But the methodology is well-designed and presented. When I read the manuscript, I could not understand, if the authors suggest to use this method. There is a major mis-conseption at the article in this point. At conclusion section, they should more clearly state.
Response:
Thank you for pointing this out. We agree that the novel technique (SANP+culture) has a lower positivity rate for Leishmania compared to traditional fine needle aspiration (FNAP) + culture. However, SANP+culture has advantages compared to FNAP+culture and necropsy+culture such as is more practical, faster and has a shorter time to positivity by culture. Regarding the time to positivity by culture, 95% of the samples collected by SANP, 87% by necropsy and 82% by FNAP were positive in the first week of culture. Therefore, our results demonstrated that SANP can be used as an alternative tool for the diagnosis of L. infantum in dogs. Following your recommendations, in order to clarify the advantages of SANP+culture and its recommendation as an alternative tool, we made changes in the title (page 1, lines 2 to 4), simple summary (page 1, lines 24, 29 to 32 and 34), abstract (page 1, lines 41 to 48) and conclusions (page 7, lines 261-265).
The article also give frequency data in dogs, the title should involve this information in addition to assessment of novel technique. The technique also seems rather an alternative of fine needle aspirate.
Response:
We agree with your comments. The title “Assessment of a Semi-Automatic Needle for Lymph Node Sampling to Diagnose Canine Visceral Leishmaniasis” was replaced with “Performance of Culture Using a Semi-Automatic Needle as a Novel Tool for Collecting Lymph Node Samples for the Diagnosis of Canine Visceral Leishmaniasis” (page 1, lines 2 to 4).
- 1. There are many abbreviations in the text, you should use less for understanding and following the article. For example, PC: culture, necroscopy: LN
Response:
In order to follow your recommendations we replaced PC with culture, LN with necropsy, IHC with immunohistochemistry and HP with histopathology throughout the text.
- Line 33. Correct “Leishmania (Leishmania) infantum “
Response:
The correction was made. We also corrected the scientific name in the simple summary (page 1, line 21).
- Line 102. “To evaluate an innovative approach of popliteal lymph node sampling for the parasitological diagnosis of CVL, the present study reports for the first time the performance of lymph node sampling by SANP when compared to samples collected by FNAP and necropsy (LN) for the diagnosis of CVL by PC compared to HP and IHC as diagnostic techniques” this sentence is not clear. Please revise it.
Response:
We rewrite the sentence in order to clarify it, as recommended. The new sentence is shown below:
Page 3, lines 107 to 109. “Evaluating an innovative approach to popliteal lymph node sampling for the parasitological diagnosis of CVL, the present study reports for the first time the performance of lymph node sampling by SANP compared to samples collected by FNAP and necropsy.”
.
Overall, the quality of English is well.
Response:
We reviewed the English language of the article as recommended.
Reviewer 2 Report
Comments and Suggestions for Authors
The study is well structurated and presents a useful tool to be considered as an additional support for Leishmaniasis diagnosis in dogs/cats.
Few suggestions are reporte in the attached file.

Author Response
Animals - Manuscript # 3344307
Response to reviewers
Reviewer 2
Comments and Suggestions for Authors
The study is well structurated and presents a useful tool to be considered as an additional support for Leishmaniasis diagnosis in dogs/cats.
Few suggestions are reported in the attached file.
Response:
Thank you for your comments that helped to improve the manuscript. Following your suggestions have replaced the text:” Parasitological tests are the reference standard for diagnosing canine visceral leishmaniasis (CVL) as they allow demonstration of the parasite, confirming the infection [1]. The parasitological tests used for the routine diagnosis of CVL are cytopathological examination, parasitological culture (PC), histopathology (HP), and immunohistochemistry (IHC).” with the text below:
Page 2, lines 61-65. “Serological tests and clinical examination are usually the first step in the diagnosis of canine visceral leishmaniasis (CVL); however, parasitological tests are the reference standard for confirming infection with Leishmania spp. since they allow demonstration of the parasite [1,4]. The most useful parasitological tests to confirm infection with Leishmania spp. during routine diagnosis of CVL are cytopathological examination, culture, histopathology, and immunohistochemistry.”
The following reference was included as recommended:
- Paltrinieri, S.; Solano-Gallego, L.; Fondati, A.; Lubas, G.; Gradoni, L.; Castagnaro, M.; Crotti, A.; Maroli, M.; Oliva, G.; Roura, X.; Zatelli, A.; Zini, E. Canine Leishmaniasis Working Group, Italian Society of Veterinarians of Companion Animals. Guidelines for diagnosis and clinical classification of leishmaniasis in dogs. J Am Vet Med Assoc. 2010, 236, 1184-1191, doi: 10.2460/javma.236.11.1184.
Therefore, we updated the number of references through the text.
Reviewer 3 Report
Comments and Suggestions for Authors
The work aims to investigate the reliability and performance of SANP (semi-automatic needle puncture) compared to FNAP (fine needle aspiration puncture) combined with culture (PC). The study is conducted on euthanized dogs.
The results show that, contrary to the Reviewer's expectations (more tissue = more leishmania = more positivity), FNAP + PC is better/similar than SANP + PC. In addition, SANP is associated with greater invasiveness and greater pain, if conducted on alive dogs (from this point of view AA should consider that diagnosis of leishmaniasis is mainly based on serological methods rather than cytological/histological evaluation).
Other aspects are highlighted by the authors, but none of these seem different from what has been widely demonstrated by previous studies. A further limitation of the work is that no information is given regarding the clinical staging of the dogs subjected to euthanasia.
Finally, it would have been appropriate to describe the culture medium used.
The feeling that one gets reading this work is that there is no real advantage in using SANP compared to FNAP, especially if the technique is to be performed on the alive animal.
Therefore, unless I have missed something, the work should be rejected in its current form and completely rewritten considering other positive aspects regarding the use of FNAP compared to SANP.
Author Response
Animals - Manuscript # 3344307
Response to reviewers
Reviewer 3
Comments and Suggestions for Authors
The work aims to investigate the reliability and performance of SANP (semi-automatic needle puncture) compared to FNAP (fine needle aspiration puncture) combined with culture (PC). The study is conducted on euthanized dogs.
The results show that, contrary to the Reviewer's expectations (more tissue = more leishmania = more positivity), FNAP + PC is better/similar than SANP + PC. In addition, SANP is associated with greater invasiveness and greater pain, if conducted on alive dogs (from this point of view AA should consider that diagnosis of leishmaniasis is mainly based on serological methods rather than cytological/histological evaluation).
Response:
Thank you for pointing this out. We agree that the novel technique (SANP+culture) has a lower positivity rate for Leishmania compared to traditional fine needle aspiration (FNAP) + culture. However, SANP+culture has advantages compared to FNAP+culture and necropsy+culture such as is more practical, faster and has a shorter time to positivity by culture. Regarding the time to positivity by culture, 95% of the samples collected by SANP, 87% by necropsy and 82% by FNAP were positive in the first week of culture. Therefore, our results demonstrated that SANP can be used as an alternative tool for the diagnosis of L. infantum in dogs. Following your recommendations, in order to clarify the advantages of SANP+culture and its recommendation as an alternative tool, we made changes in the title (page 1, lines 2 to 4), simple summary (page 1, lines 24, 29 to 32 and 34), abstract (page 1, lines 41 to 48) and conclusions (page 7, lines 261-265).
Serological tests and clinical examination are usually considered the first step for diagnosing canine visceral leishmaniasis (CVL), but parasitological tests are the reference standard for confirming the infection with Leishmania spp. as they allow demonstration of the parasite. The combination of several diagnostic methods is the best approach to detecting this infection [25].
25-Teixeira, A.I.P.; Silva, D.M.; Vital, T.; Nitz, N.; de Carvalho, B.C.; Hecht, M.; Oliveira, D.; Oliveira, E.; Rabello, A.; Romero, G.A.S. Improving the reference standard for the diagnosis of canine visceral leishmaniasis: a challenge for current and future tests. Mem. Inst. Oswaldo Cruz 2019, 114, doi:10.1590/0074-02760180452
In order to better clarify the importance of serology and parasitological tests for the diagnosis of canine visceral leishmaniasis we have replaced the following text in the introduction:” Parasitological tests are the reference standard for diagnosing canine visceral leishmaniasis (CVL) as they allow demonstration of the parasite, confirming the infection [1]. The parasitological tests used for the routine diagnosis of CVL are cytopathological examination, parasitological culture (PC), histopathology (HP), and immunohistochemistry (IHC).” with the text below:
Page 2, lines 61-65. “Serological tests and clinical examination are usually the first step in the diagnosis of canine visceral leishmaniasis (CVL); however, parasitological tests are the reference standard for confirming infection with Leishmania spp. since they allow demonstration of the parasite [1,4]. The most useful parasitological tests to confirm infection with Leishmania spp. during routine diagnosis of CVL are cytopathological examination, culture, histopathology, and immunohistochemistry.”
The following reference was included as recommended:
- Paltrinieri, S.; Solano-Gallego, L.; Fondati, A.; Lubas, G.; Gradoni, L.; Castagnaro, M.; Crotti, A.; Maroli, M.; Oliva, G.; Roura, X.; Zatelli, A.; Zini, E. Canine Leishmaniasis Working Group, Italian Society of Veterinarians of Companion Animals. Guidelines for diagnosis and clinical classification of leishmaniasis in dogs. J Am Vet Med Assoc. 2010, 236, 1184-1191, doi: 10.2460/javma.236.11.1184.
Therefore, we updated the number of references through the text.
Other aspects are highlighted by the authors, but none of these seem different from what has been widely demonstrated by previous studies. A further limitation of the work is that no information is given regarding the clinical staging of the dogs subjected to euthanasia.
Response:
In order to clarify the advantages of SANP of popliteal lymph nodes combined with culture and its recommendation as an alternative tool for the diagnosis of canine visceral leishmaniosis, we made changes as described above.
Following you comment about the limitation of the study we added the following phrase in the results (page 4, lines 167 to 168): “A limitation of this study is that clinical staging of the dogs was not performed.”
Finally, it would have been appropriate to describe the culture medium used.
Response:
The culture medium used was Novy-MacNeal-Nicolle + Schneider’s biphasic media, supplemented with antibiotic and antifungal agents. This information was included in the material and methods section (page 3, lines 128 to 129), as recommended.
The feeling that one gets reading this work is that there is no real advantage in using SANP compared to FNAP, especially if the technique is to be performed on the alive animal.
Therefore, unless I have missed something, the work should be rejected in its current form and completely rewritten considering other positive aspects regarding the use of FNAP compared to SANP.
Response:
The SANP+culture has advantages compared to FNAP+culture and necropsy+culture such as is more practical, faster and has a shorter time to positivity by culture as explained above. In order to clarify the advantages of SANP+culture and its recommendation as an alternative tool for the diagnosis of L. infantum in dogs, we made changes in the title (page 1, lines 2 to 4), simple summary (page 1, lines 24, 29 to 32 and 34), abstract (page 1, lines 41 to 48) and conclusions (page 7, lines 261-265).
Round 2
Reviewer 3 Report
Comments and Suggestions for Authors
Tha paper may be accepted for publication